# Vascular Occlusion in Kidney Biopsy Is Characteristic of Clinically Manifesting Thrombotic Microangiopathy

**DOI:** 10.3390/jcm11113124

**Published:** 2022-05-31

**Authors:** Marja Kovala, Minna Seppälä, Kati Kaartinen, Seppo Meri, Eero Honkanen, Anne Räisänen-Sokolowski

**Affiliations:** 1Department of Pathology, Helsinki University Hospital and Helsinki University, 00029 Helsinki, Finland; anne.raisanen-sokolowski@hus.fi; 2Department of Nephrology, Helsinki University Hospital and Helsinki University, 00029 Helsinki, Finland; minna.seppala@hus.fi (M.S.); kati.kaartinen@hus.fi (K.K.); eero.honkanen@fimnet.fi (E.H.); 3Department of Bacteriology and Immunology, Helsinki University Hospital and Helsinki University, 00029 Helsinki, Finland; seppo.meri@helsinki.fi

**Keywords:** thrombotic microangiopathy, histological TMA, clinical TMA, kidney biopsy, secondary HUS, atypical HUS

## Abstract

Thrombotic microangiopathy (TMA) can sometimes manifest only histologically. Our aim was to retrospectively compare biopsy-proven adult TMA patients showing only histological (h-TMA) or both histological and clinical (c-TMA) TMA in 2006–2017. All native kidney biopsies with TMA were included. Biopsies were re-evaluated by light and electron microscopy, and immunofluorescence. Clinical characteristics, laboratory variables, and treatments were recorded from the electronic medical database. Patients were categorized into h-TMA and c-TMA and these groups were compared. In total, 30 biopsy-proven cases among 7943 kidney biopsies were identified and, of these, 15 had h-TMA and 15 c-TMA. Mean follow-up was 6.3 y, and 73.3% had secondary hemolytic uremic syndrome (HUS) and the rest were atypical HUS. Patient characteristics, treatments, and kidney, and patient survival in the groups were similar. Statistically significant differences were found in histological variables. Vascular myxoid swelling and vascular onion-skinning were almost exclusively detected in c-TMA and, thus, vascular occlusive changes indicate clinically apparent rather than merely histological TMA. In addition, regardless of clinical presentation, kidney and patient survival times were similar in the patient groups highlighting the importance of a kidney biopsy in the case of any kidney-related symptoms.

## 1. Introduction

Thrombotic microangiopathy (TMA) is a pathologic description of microvascular occlusive disorder leading to characteristic histological lesions of thrombus formation, and vascular endothelial injury [1]. Clinically, TMA has been characterized by a classic triad of thrombocytopenia, microangiopathic hemolytic anemia, and an organ injury [2], but nowadays asymptomatic presentation without clinically apparent TMA has been described as well, where TMA is only detected in a kidney biopsy [3,4].

TMA can show a variety of different histological lesions, depending on the activity and timing of the disease. Active lesions include fragmented red blood cells, endothelial swelling, or occlusion of glomeruli or arterioles, mesangiolysis, or microaneurysms in glomeruli, whereas chronic lesions include double contours in glomerular basement membranes (GBM), and fibrous intimal thickening (“onion-skinning”) of arteries [2]. As chronic changes can eventually lead to a formation of lobulated glomeruli, there is a possibility of misdiagnosing TMA and membranoproliferative glomerulonephritis (MPGN) [5]. In addition, these entities have been intermingled clinically [6]. In TMA, however, the immunofluorescence (IF) findings are completely, or mainly, negative, because complement activation happens on cell surfaces [7] as opposed to a large-scale fluid-phase activation in MPGN [8]. Biopsies from TMA patients can still sometimes show unspecific entrapment of IgM and C3 and, also, immune complexes, especially if the condition is associated with connective tissue disease [9]. In addition, concurrent and consecutive C3 glomerulopathy (C3G) and atypical hemolytic uremic syndrome (aHUS) have been reported. In selected cases, the location of factor H mutation may play a crucial role in differentiating these entities pathophysiologically [10]. In aHUS, factor H mutation is usually located at the C-terminus (short consensus repeats 19–20), which mediates cell-surface protection [7] and in C3G at the N-terminus [11].

Some reports have shown that TMA-associated glomerular and vascular changes are more frequent in certain TMA patient groups. Neto et al. observed that microthrombi were more frequent in aHUS than in enterohemorrhagic *E. coli* (EHEC) Shiga toxin mediated HUS (STEC-HUS), while ischemia was more common in the latter [12]. Yu et al. observed that acute vascular changes were more common in aHUS than in malignant hypertension or pregnancy-associated HUS, whereas chronic and arterial changes were more common in malignant hypertension [13]. However, histological lesions are heterogenous and no specific lesions have been found to be tightly associated with these etiologies of TMA [12,13]. TMA can also present without formation of thrombi, and it has been suggested to be called microangiopathy without thrombosis [2]. Histologically, TMA can also include antibody-mediated rejection seen on transplant biopsies [14].

Clinically, TMA can present with heterogenous symptoms, such as acute kidney injury, neurological abnormalities, gastrointestinal symptoms, or purpura. Patients are often critically ill, but there are also patients who have a more benign, or even asymptomatic presentation [3]. TMA can affect multiple organs, but kidneys are most often affected [13]. Often a secondary trigger is required for the clinical manifestation of TMA [10].

Traditionally, TMA has been classified into thrombotic thrombocytopenic purpura (TTP), based on severe ADAMTS13 deficiency, typical hemolytic-uremic syndrome (HUS) based on a Shiga toxin-producing pathogen, secondary HUS (sHUS), which is caused by multiple factors, including medications, infections, malignancies, organ transplantations, connective tissue diseases, and, nowadays, COVID-19, etc., and into aHUS for complement dysregulation-mediated HUS [10,15]. Most cases of TTP are linked to the formation of autoantibody against disintegrin and metalloproteinase with a thrombospondin type 1 motif, member 13 (ADAMTS13) and the clinical scenario is that of minimal kidney but various degree of neurological symptoms [10,16]. Thrombocytopenia is usually severe in TTP [16]. In contrast, aHUS has predominant kidney derangements and less other involvement [16]. Also the level of platelets is higher in aHUS than in TTP [10]. The treatment of TMA is determined by the etiology of the disease. Plasma exchange, immunosuppressive therapy, and anti-complement medications are the cornerstones of the treatment in primary TMAs.

The aim of this study was to gain deeper understanding on the histological spectrum of native kidney TMA and the characteristics of these patients, regardless of their clinical presentation. The goal was also to analyze features of clinical presentation as opposed to a mere histological diagnosis, effects of a given treatment and to find possible associations between histology and prognosis.

## 2. Materials and Methods

### 2.1. Study Population

This was a retrospective, single-center study of Helsinki University Hospital, a district of Finland covering about 1.7 million inhabitants (approximately 30% of the total Finnish population). All adult patients over 18 y of age with native kidney biopsy proven TMA or similar histological changes between the years 2006–2017 were included. All patients were divided into clinical TMA (c-TMA), or histologic TMA (h-TMA) groups based on whether they had also clinically evident TMA disease or not. C-TMA had typical TMA findings in peripheral blood, whereas h-TMA had no anemia, thrombocytopenia, red blood cell fragments, or abnormal hemolytic tests at the time of biopsy-proven diagnosis. We also categorized the patients based on the clinical presentation to TTP, aHUS, sHUS and typical STEC-HUS.

Identification of the patients was carried out from the database of the Department of Pathology yielding a total of 7.943 kidney biopsies during 2006–2017. Search criteria were TMA, thrombus, and malignant hypertension, with all their synonyms and abbreviations from the pathologic-anatomic diagnosis and the report. MPGN, its pattern of injury, and C3G were also re-evaluated to detect any misdiagnosed cases. Exclusion criteria were insufficient biopsy material for re-evaluation and primary MPGN diagnosis or pattern of injury resulting from another underlying disease, such as IgA nephropathy or systemic lupus erythematosus. The eligibility and clinical diagnosis of problematic cases was judged independently by two nephrologists having expertise in TMA. Study was reviewed by the hospital’s Ethical Committee (HUS/2520/2018) and research permit (HUS/459/2018) was granted. The study was conducted according to the Declaration of Helsinki.

Electronic medical records were reviewed for family history, age, and sex distribution, possible triggering factors, clinical information including laboratory values and therapies employed at the onset of the disease and during follow-up. The follow-up time started from the time of diagnostic biopsy (2006–2017) until the patient’s death or until the end of year 2019.

ADAMTS13-activity test was used to identify patients with TTP. If ADAMTS13 was not assessed, creatinine levels and platelet count were utilized to determine the probability of severe ADAMTS13 deficiency, as Coppo et al. established in 2010 [17]. If serum creatinine level was ≤200 µmol/L and platelet count ≤30 × 10^9^/L at diagnosis, the patient was considered to have TTP. Typical HUS was identified with stool samples positive for Shiga toxin-producing bacteria. If stool samples were not available, typical HUS was excluded based on clinical grounds. For the remaining patients, we determined whether there existed any known cause for sHUS, such as drugs associated with TMA, transplantation, other infections (non-Shiga toxin-producing pathogens), autoimmune diseases, malignancies, cobalamin C deficiency, myeloproliferative diseases, or malignant hypertension [18]. If a complement pathway disturbance related to target recognition was discovered, the patient was classified as having aHUS. If no cause for sHUS was identified, the patient was labelled as having aHUS even if complement genetics or autoantibody tests were negative. If there was any doubt as to how the patient should be classified, we reviewed the case among members of the research team.

Progressive disease was defined as the need of kidney-replacement therapy during the follow-up. Classic triad was defined as microangiopathic hemolytic anemia, thrombocytopenia, and kidney manifestation. Increased serum creatinine concentration and decreased estimated glomerular filtration rate (eGFR), or hematuria and/or proteinuria were observed as a sign of kidney manifestation.

Primary outcomes were to compare clinical, laboratory, and histological characteristics between c-TMA and h-TMA. Secondary outcomes were to compare kidney, and overall prognosis as well as to compare prescribed treatments between c-TMA and h-TMA.

### 2.2. Laboratory Analyses

Laboratory variables were analyzed at the accredited Helsinki University Hospital laboratory (HUSLAB) using standard chemical methods. Kidney function was measured and estimated using serum creatinine, for which upper normal values were 100 μmol/L in males and 90 μmol/L in females, and eGFR calculated according to the Chronic Kidney Disease Epidemiology Collaboration equation (CKD-EPI), for which cut-off for lower normal value was 60 mL/min/1.73 m^2^. Complement analyses were performed using the Wieslab complement system screen method (activity percentages of serum classical, alternative and lectin pathways, S-CH100Cl, S-CH100Al, and S-CH100L, respectively) and nephelometrically measuring serum levels of complement C3 and C4, (S-C3, and S-C4, respectively). C3 nephritic factor (C3Nef) was analyzed by in-house immunofixation electrophoresis, in which the ability of the patient’s serum to activate the alternative pathway of complement in normal serum in the presence of magnesium ethylene glycol tetra-acetic acid (MgEGTA) was examined.

Genetic analyses were performed utilizing aHUS mutation panel (9 gene panel including ADAMTS13, C3, membrane cofactor protein (MCP), complement factor B, complement factor H, complement factor H related protein 5, complement factor I, diacylglycerol kinase epsilon (DGKE), and thrombomodulin) at Blueprint Genetics laboratory, Helsinki, Finland.

Factor H-antibodies were quantified using an ELISA assay with the Helsinki protocol [19]. Human sera for a negative and positive control were included as controls. To quantify anti-factor H antibodies Nunc Maxisorp plates were first coated with 100 μL portions of factor H (5 µg/mL). After an overnight incubation at 4 °C, the plates were washed with phosphate-buffered saline (PBS) containing 0.05% Tween 20 and blocked with 200 μL of PBS-Tween at ambient temperature for 2 h. Samples were analyzed in duplicate. Serum samples were diluted 1/20 and 80 μL portions were added and incubated for 2 h at 37 °C, whereafter the plates were washed with PBS. Anti-human IgG-HRP (horse radish peroxidase) diluted at 1:2000 in PBS was added. After an incubation at 37 °C for 1 h, the plates were washed with PBS and o-phenylenediamine dihydrochloride (OPD) substrate was added. The reaction was stopped with 120 μL of 0.5 M H_2_SO_4_, and a spectrophotometer was used to measure the optical density of samples at 492 nm wavelength.

Factor H and the factor H-related proteins were analyzed by immunoblotting having normal human serum as a control. Into a 4–12% SDS-PAGE gel, appropriately diluted samples were added at 1:100 and 1:300 dilutions. The samples were then run for 45 min at 165 V in 1× MES buffer, whereafter they were transferred onto a filter membrane. Non-specific binding sites were blocked for 1 h at room temperature with 1 mL of 5% non-fat dry milk prepared in 1× PBS and 0.05% Tween 20. Later, primary goat anti-factor H antibody was added into the solution and incubated overnight at 4 °C. The membranes were then washed at room temperature for 1 h with 15–20 min changing interval of 2–3 mL 1× PBS and 0.05% Tween 20. Secondary rabbit-anti-goat HRP-conjugated antibody was added with 5% non-fat dry milk and incubated at room temperature for 1 h. The washing with PBS changing intervals was then repeated at room temperature for 1 h. The bands were then visualized with an in-house protocol for electrochemiluminescence recipe for 1 min at room temperature. Films were then developed at different exposure times.

### 2.3. Pathological Analyses

Kidney biopsy was re-evaluated by a nephropathologist (AR-S) together with, at that time, the pathology resident in training (MK). The final histological diagnosis was defined by the nephropathologist blinded to the clinical diagnosis.

The light microscopy (LM) sections were fixed with formaldehyde, after which 2 µm paraffin sections were stained with Periodic-Acid-Schiff, hematoxylin-eosin, Jones silver stains, Congo red stain, and, depending on the availability of the stain, Masson’s trichrome stain. LM findings were classified according to the Definition of Glomerular Lesion by Renal Pathology Society [20] and 2018 Banff classification [21] and biopsies were given a final histological diagnosis of acute, subacute, and chronic TMA. Acute changes of TMA included mesangiolysis, glomerular or arteriolar thrombus, microaneurysms, glomerular or arteriolar necrosis, crescent formation, and arteriolar myxoid intimal swelling and chronic TMA changes included doubling of the GBM, mesangial matrix expansion, lobulated pattern of glomeruli, and intimal onion-skinning [8,11]. However, the final histological diagnosis was judged visually implementing the knowledge of acute and chronic changes mentioned above. Subacute diagnosis was rendered if the biopsy included both acute and chronic changes.

Immunofluorescence (IF) staining results were obtained from the previously written pathology reports and photographs. If there were any discrepancies, the staining was reperformed on frozen sections (F-IF), which were stained with goat-anti-human IgG, IgA, IgM, C3, C1q, and kappa antibodies and burro-anti-human for lambda antibody (Ventana Laboratories, Tucson, AZ, USA). The findings were graded 0–4, according to the intensity of positive staining, where 0 was negative, 1 trace, 2 mild, 3 moderate, and 4 strong. Paraffin-section IF (P-IF) after proteinase K retrieval of antigen [22] was performed on four biopsies as there was not sufficient material for reperforming the F-IF and they were graded similarly.

Electron microscopy (EM) findings were obtained from the original pathology reports and photographs and were re-evaluated according to previously mentioned criteria of consensus report [20]. If EM was not performed formerly and there was a previously acquired glutaraldehyde-fixed sample available, it was processed to sections for transmission EM. EM analysis was performed for a total of 24 (80%) biopsies and the findings were recorded for following variables: number of glomeruli, wrinkled GBM, thickness of GBM, subendothelial widening of GBM, doubled GBM, enlarged glomerular endothelial cells, loss of fenestration on endothelium of GBM, podocyte enlargement, amount of podocyte foot processes effacement, podocyte microvillous transformation, glomerular intracapillary inflammatory cells, mesangial expansion, subendothelial deposits, subepithelial deposits, intramembranous deposits, mesangial deposits, and deposit substructure.

### 2.4. Statistical Analyses

Statistical analysis was completed by a professional statistician using R software version 4.0.4 (R Core Team, Vienna, Austria, 2021). T-test was used for assessing the differences in means, test of proportions (Fischer’s exact test) for assessing the differences between proportions, and Mann–Whitney U test for assessing the differences in medians. As the sample size or the number of values of interest was in many cases low, Fischer’s exact test was chosen instead of z-test, which is based on the normal approximation. In addition, the estimation of factors influencing the changes in binary dependent variables was carried out using binary logistic regression. The significance level of the statistical tests mentioned above was set to 0.05.

## 3. Results

### 3.1. Patient Population and Clinical Findings

The study flow of the research is depicted in Figure 1. A total of 30 TMA patients were included in the study. TMA was found in 0.4% of all adult biopsies during the years 2006–2017. The annual relative frequency of all native TMA diagnoses in our biopsy database was 0.2–0.8% during the study period. In total, 15 (50%) patients were diagnosed as c-TMA, whereas another 15 (50%) patients were diagnosed with only h-TMA. Based on all the available information, 21 (70%) of these patients were assigned as sHUS and 9 (30%) patients as aHUS. There were no patients with TTP or typical HUS.

Clinical characteristics at the time of diagnostic kidney biopsy are summarized in Table 1. Overall, there were 18/30 (60%) males and the mean age at the onset of the disease for the whole study population was 52.8 (range 24–78) y. The duration of kidney findings was estimated from the start of kidney-related symptoms (proteinuria, hematuria, decreased kidney function, or a combination of these) in h-TMA group, or from the start of laboratory kidney derangements at the time of acute illness in c-TMA group. The duration of kidney findings varied between 0–228 (mean 14.4) m in the whole study population. Of the whole study population, one patient had a family member with a history of TMA and this patient also had a disease-causing heterozygous mutation of factor H. Total 4 (13.3%) patients had a family history of an unknown kidney disease, of which one patient had a heterozygous mutation in the C3 gene and a heterozygous risk haplotype in MCP. One patient had a heterozygous risk haplotype in factor H and a heterozygous risk haplotype in MCP (data not shown).

### 3.2. Predisposing and Causative Factors of Atypical and Secondary HUS

Predisposing and causative factors and differential diagnostics of aHUS and sHUS are shown in Table 2. A predisposing factor was identified only in c-TMA group and in total for 4/9 (44.4%) of aHUS patients. These were a recent infection in two, surgery in one and multiple predisposing factors in one. Secondary HUS was most often related to multiple factors (10/30, 33.3%), including infection, autoimmune disease, malignant disease/chemotherapy, cobalamin C deficiency, medication, hypertension, organ transplantation, essential thrombocytosis, and monoclonal gammopathy of unknown significance (MGUS). The most common single reason was hypertension (5/30, 16.7%). ADAMTS13 activity was measured in 13/29 (44.8%) patients to rule out TTP. Typical HUS was ruled out in 14 (46.7%) patients by examining by PCR the presence of the EHEC Shiga toxin gene in feces samples.

### 3.3. Laboratory Variables and Paraprotein Findings

Variables indicative of hemolysis are shown in Table 3. As expected, the c-TMA group had a significantly lower hemoglobin level (103.5 ± 17.6 g/L) than the h-TMA group (119.5 ± 14 g/L) (*p* = 0.01). Lactate dehydrogenase (LDH) and erythrocyte-schistocyte levels were significantly higher in c-TMA than in h-TMA (348.4 ± 156.2 vs. 206.9 ± 40.4, *p* = 0.004 and 2.0 ± 1.5 vs. 0.1 ± 0.2, *p* = 0.002, respectively).

Complement variables along with paraprotein findings are summarized in Table 4. There were no significant differences between the groups. Plasma levels of factor B, C4d, C3d, and factor Bb were measured from one c-TMA and one h-TMA patient only, and they were all within the reference values (data not shown). The plasma levels of the soluble terminal complement complex SC5b-9 (ref. <366 ng/mL) were abnormally high in all the four c-TMA patients (650.6 ± 390.3 ng/mL), in whom it was measured. P-SC5b-9 was measured only in one h-TMA patient, in whom it was in the normal range (298 ng/mL). Functional complement activity analysis was performed for 12 patients. Decreased activities below the reference range in the lectin and classical pathway were found in one h-TMA and in two c-TMA patients. No patient had decreased or increased activities of the alternative pathway.

Paraproteins were analyzed for seven patients, and in 5/7 (71.4%) (altogether 5/30, 17%) a paraprotein either in urine or in serum was observed. It was observed in blood plasma in three patients, and in two patients the paraprotein was found both in blood, and urine. Serum IgG-kappa and IgG-lambda were the most common paraprotein types. Both were observed in two patients each. The fifth patient had both IgG-kappa and IgM-kappa in the plasma sample. Hematological disease was classified as MGUS/monoclonal gammopathy of renal significance (MGRS) in 4 of the 5 patients. An overt malignancy occurred in the fifth patient.

### 3.4. Histological Findings

In Figure 2 various morphological changes of TMA are demonstrated.

Histological findings are summarized in Table 5. Myxoid swelling was observed in 9/15 (60%) c-TMA and only in one (6.7%) h-TMA patient (*p* = 0.005). Onion-skinning was seen in 6/15 (40%) c-TMA patients, but not in any h-TMA patients (*p* = 0.017). Vascular intramural fibrin and vascular thrombus formation were more common in the c-TMA group than in the h-TMA group, but the differences were not statistically significant. The degree of arterial sclerosis was similar in both groups.

F-IF or PIF was available for all biopsies to verify the diagnosis. Traces (1+) or weak (2+) positivity of immunoglobulins, C3 and/or C1q were noted in few patients. No biopsy showed moderate or strong (3+–4+) IF staining.

By EM, loss of endothelial fenestration was more often found in c-TMA cases. In addition to the observed features of TMA, thin basement membrane disease was observed in two (20%) h-TMA patients and two (15.4%) c-TMA patients (Table 5).

In addition, the timeframe in which the patient was biopsied in comparison to the onset of clinical symptoms (c-TMA), or from kidney-related symptoms (h-TMA) was analyzed between groups, which did not show statistically significant differences between the patient groups.

### 3.5. Treatment, Renal and Overall Prognosis

Plasma exchange at the time of histological diagnosis was used for 6 (40%) c-TMA patients, but none of h-TMA patients (*p* = 0.017). Acute dialysis was needed for 6 (40%) of c-TMA patients and for 1 (6.7%) h-TMA patient (*p* = 0.08), immunosuppressive medication for 8 (53.3%) and 3 (20%) patients, respectively. Blood pressure medication alone was used for one (6.7%) patient both in c-TMA and h-TMA groups.

The mean follow-up time from a kidney biopsy was not significantly different between the groups being 6.9 y for h-TMA and 5.6 y for c-TMA (Table 6). There were no significant differences between c-TMA and h-TMA patients in the need for chronic dialysis, kidney transplantation, or abnormal kidney function not necessitating kidney replacement therapy. Kidney failure resulting in dialysis or transplantation did not differ between the groups (Figure 3). The survival probability of h-TMA patients seemed to be poorer (*p* = n.s.), but this might be explained by the fact that of the five patients who died during the follow-up, four were h-TMA (two had cancer-related TMA and two hypertension-related TMA), and only one c-TMA (cancer treatment- related TMA) (Figure 4).

In univariate analysis of the whole patient group, severe doubled GBM, glomerular thrombi, mesangiolysis, arterial myxoid intimal swelling, arterial onion-skinning, and degree of podocyte foot process effacement were statistically significant determinants of progressive kidney disease. However, in multivariate analysis none of these factors were significant (data shown in the Appendix A).

## 4. Discussion

To our knowledge, we have demonstrated for the first time that occlusive vascular changes seem to be almost exclusively found in also clinically manifest forms of TMA (c-TMA), whereas in only h-TMA the vascular changes are rare. The occlusive changes cause acute ischemic injury to the glomeruli. Thus, kidney function deteriorates causing clinical symptoms and signs of c-TMA. Predisposing factors of aHUS were only observed in c-TMA, but not in h-TMA group. This might merely reflect the fact that since the precise origin of TMA was difficult to know for certain in h-TMA patients, the potential predisposing or triggering factors were infeasible to reveal, too. Interestingly, our study showed that c-TMA and h-TMA have surprisingly similar kidney and patient survival. This indicates that also h-TMA is a serious pathological condition leading to an abnormal kidney function, despite lacking clinical symptoms of TMA.

Most TMA patients in the clinical routine practice do not undergo a kidney biopsy, but rather the diagnosis is based on the clinical laboratory variables. TTP patients usually have no or only minimal kidney disease abnormalities. Therefore, they are virtually never biopsied. All HUS patients, on the contrary, have various degrees of kidney abnormalities either in kidney function and/or urine tests. However, most of HUS patients are anyhow treated without histological confirmation. Diagnosis of TMA does not require histological confirmation. Clinically the patient is often critically ill and, therefore, undergo kidney biopsy only rarely. These reasons most likely explain the low number of biopsy-proven TMA cases (0.4%) in our extensive material of more than 7000 kidney biopsies. A nationwide cross-sectional analysis from Japanese kidney biopsy registry showed that 0.39% had a diagnosis of TMA of various causes [1]. Also, a study from China reported TMA to be found in 1.4% of all kidney biopsies [13]. However, no information was available on the percentage of isolated kidney TMA in either of these studies. Thus, the number of TMA patients in our study is comparable to what has been published elsewhere.

Based on the literature it is unclear, if biopsy-proven TMA showing only histological features shares the same underlying etiology, prognosis, and clinical course than TMA with also clinical features. Most reports have compared different TMA subtypes (secondary HUS of various etiology, aHUS, typical HUS) [23,24] or reported findings in one subtype only [25], but none have reported comparisons based on the clinical scenario (h-TMA versus c-TMA). In a French retrospective report of 110 secondary HUS patients, kidney biopsy was performed in 46% of the patients. However, it was not reported how many were only histological diagnoses [23]. In an Indian report of 33 native kidney TMA patients some hypertension and postpartum TMAs were reported to have only isolated arterial changes, but no further information was available [24]. In our report, it was evident that despite a somewhat better renal function in h-TMA patients at onset of the disease, kidney prognosis was, nevertheless, as poor in h-TMA as in c-TMA, because 27% of h-TMA patients needed chronic dialysis during follow-up compared to 33% in c-TMA. The frequency of kidney transplantation was not different between our groups either, being 14% and 11% in h-TMA and c-TMA, respectively. Kidney failure occurred in 37% of sHUS and in 71% of aHUS patients in the French report [23]. Interestingly, in our study 48% (10/21) of sHUS and 56% (5/9) of aHUS patients had only histological TMA.

It is not known what the optimal treatment for h-TMA patients is, and whether the treatment should be adjusted according to the clinical presentation of the disease. Should aHUS presenting as h-TMA be treated differently than aHUS presenting as c-TMA? The same dilemma applies to sHUS, as well, although sHUS is more heterogenous in terms of etiology. In our study, none of the h-TMA patients received anti-complement medication, or plasma exchange therapy compared to 66.7%, and 40% of the c-TMA patients, respectively. Similarly, only 20% of the h-TMA patients received any form of immunosuppressive therapy compared to 53% in the c-TMA group.

TMA histology comprises several different characteristics and the changes seen by LM and EM can be thought to result from continuum of time and the stage as well as the severity of the disease. In the very early phases of the disease the changes may be evident only as subendothelial widening evident only in EM proceeding later to changes visible in LM. As the overall biopsy morphology was divided into acute, subacute, and chronic changes, our study showed that in the whole study population the incidence of chronic changes increased substantially after 12 w from the onset of disease. There were no statistical differences as we compared these between the patient groups.

In our study vascular occlusive changes were almost exclusively linked to c-TMA as myxoid swelling was observed in 60% and onion-skinning in 40% of the c-TMA patients. Vascular intramural fibrin and thrombus formation were also more common in c-TMA group, although not statistically significantly. This might be explained by occlusive vascular changes causing ischemic glomerular injury and, thus, more severe functional kidney impairment at the onset of the disease. Furthermore, it can be speculated that smoldering endothelial injury causes remodeling of GBM leading to a milder clinical presentation and subclinical TMA. A Chinese study observed that acute changes were more common in aHUS than in conditions with malignant hypertension or pregnancy-related HUS, whereas chronic and arterial changes were more common in malignant hypertension [13]. However, histological lesions were heterogenous and no specific lesions were observed to be tightly associated with different etiologies of TMA [12,13]. In our study, the vascular occlusive changes were linked to clinically evident disease despite of the etiology or the time interval, in which the biopsy was acquired after the onset of symptoms. We did not observe glomerular or tubulointerstitial changes to be different in the patient groups nor were there differences in the kidney or patient survival between the groups. It is not clear why h-TMA patients anyhow later developed significantly reduced kidney survival.

TMA has a provisional status among various MGRS diagnoses, but the mechanisms of monoclonal gammopathy-associated TMA are not fully understood [26]. In a series of 146 TMA patients, a monoclonal gammopathy was present in 14% [27]. This number is close to that observed in our study (17%).

Considering the ultrarare nature of all TMA diagnoses and the fact that only a minority will ever be biopsied during their illness, our study represents a relatively large one center analysis of TMA patients. However, our study has a few limitations as well, as it was a single-center study, generalizability to a broader population is limited. Moreover, it covers only sHUS and aHUS and, therefore, it cannot be applied to all TMA patients. C-TMA patients with a kidney biopsy during the acute phase of the disease are highly selected and may not fully represent the histological map of c-TMA patients. The limited number of TMA patients affects the size of the study population and, thus, the validity of multivariate logistic regression carried out is restricted. The retrospective nature of our study may also hinder firm conclusions and generalizability, and not all patients had all variables of interests analyzed.

## 5. Conclusions

In conclusion, our study indicates that there is a large number of cases with smoldering endothelial injury causing renal TMA that has similar patient and kidney survival despite the clinical presentation. The comparison between h-TMA and c-TMA implies that more attention should be paid on h-TMA patients despite the etiology of their TMA. These patients might benefit on more aggressive treatments than currently used. However, further studies focusing on various aspects on h-TMA are needed.

## Figures and Tables

**Figure 1 jcm-11-03124-f001:**
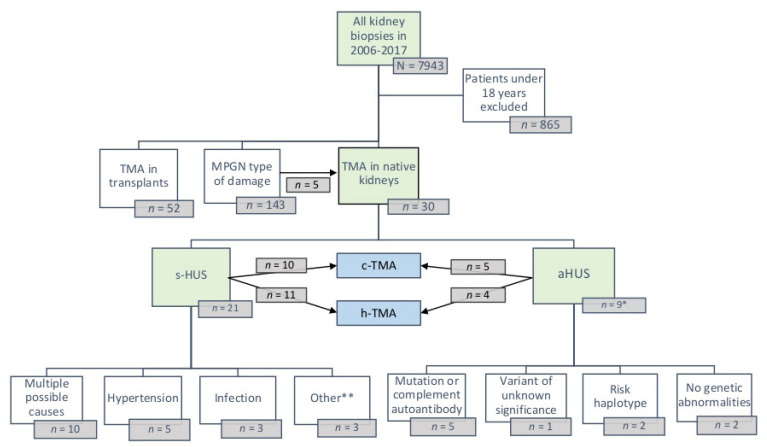
Study flow of the research. * Some patients are categorized twice based on the complement finding, ** malignant disease, and rheumatic disease.

**Figure 2 jcm-11-03124-f002:**
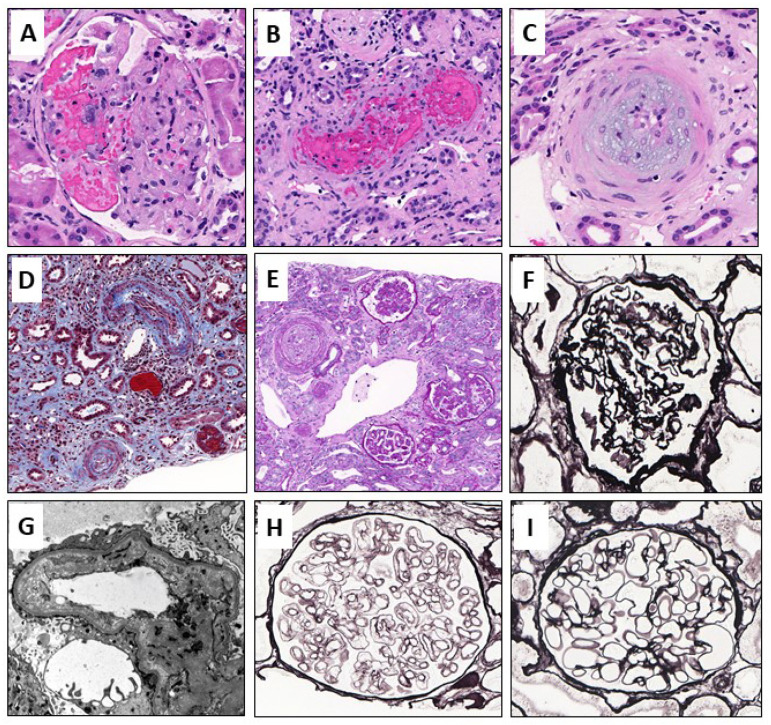
Illustrations of acute, subacute, and chronic TMA changes: (**A**) In acute and active TMA can be seen microaneurysms and intracapillary thrombi, although they are rather rare; (**B**) Thrombi can be seen also in the small arterioles; (**C**) Often there is endothelial injury that causes accumulation of fluffy, myxoid material underneath the endothelium causing occlusion of the vessel; (**D**) Eventually, the vascular injury becomes organized and forms “onion skin”-like appearance. Surrounding interstitial tissue becomes scarred and tubular epithelium is injured; (**E**,**F**) Vessel occlusion causes ischemic injury to the glomeruli. The basement membrane is wrinkled; (**G**) Electron micrograph of the capillary loop that has endothelial injury shown as electron-lucent material underneath endothelium; (**H**) Ongoing, smoldering endothelial injury causes remodeling of the basement membrane and duplicates become visible even in light microscopy; (**I**) Normal glomerulus with thin and intact basement membranes for comparison to previous ones. Original stainings and magnifications: (**A**–**C**) Hematoxylin-eosin and 400×, 200×, and 400×, respectively, (**D**) Masson Trichrome 200×, (**E**) Periodic-Acid-Schiff 200×, (**F**,**H**,**I**) Jones Methenamine Silver, 400× for all, (**G**) uranyl acetate 12,000×.

**Figure 3 jcm-11-03124-f003:**
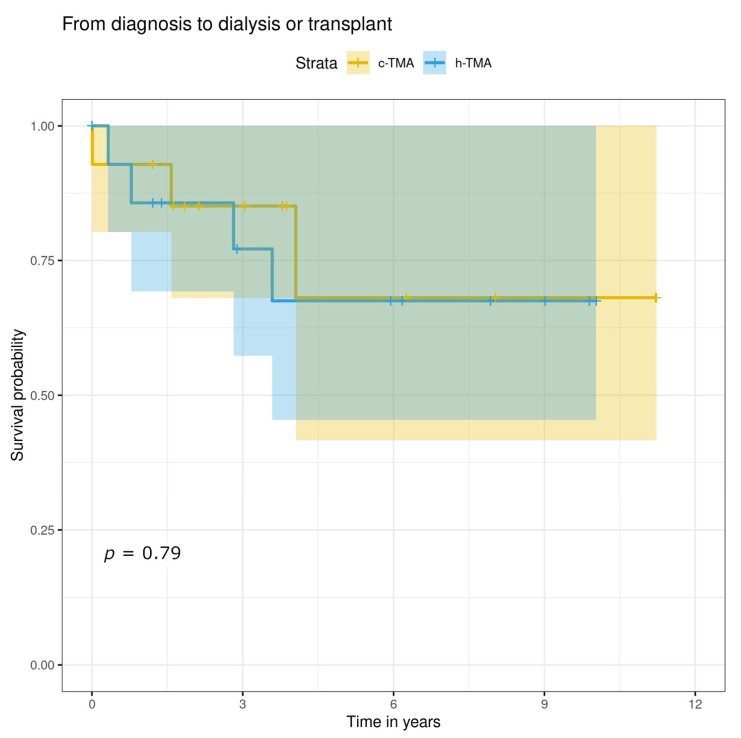
Kaplan–Meier curve on kidney failure from diagnosis resulting in dialysis or transplantation with c-TMA and h-TMA patients.

**Figure 4 jcm-11-03124-f004:**
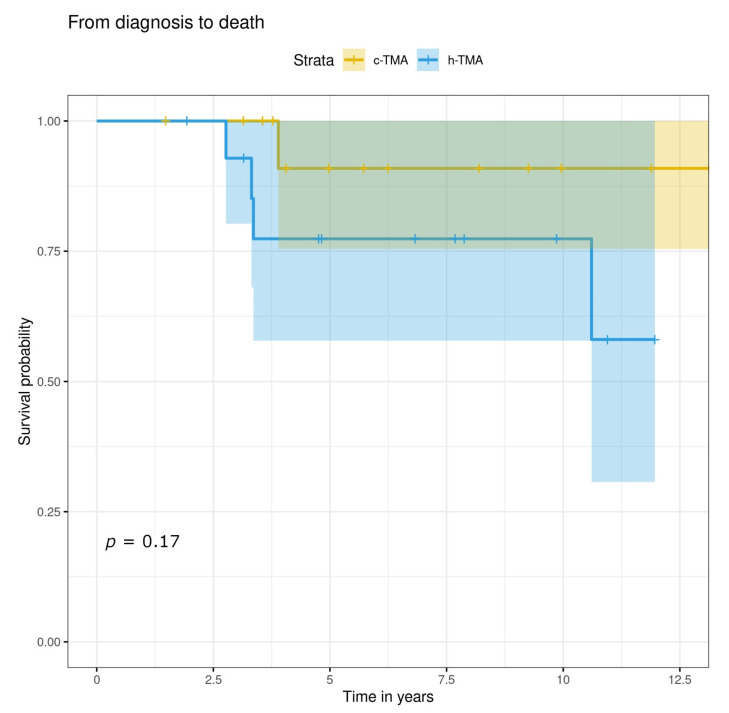
Kaplan–Meier curve on patient survival from diagnosis to death with c-TMA and h-TMA patients.

**Table 1 jcm-11-03124-t001:** Clinical characteristics of the patients recorded at the time of diagnosis. Values are expressed as means (range in parenthesis), or as number of patients (percentages). *p*-values shows the comparison between h-TMA and c-TMA groups.

Baseline Variable	h-TMA *n* = 15	c-TMA *n* = 15	*p*-Value
Age (y, range) ^1^	56.7 (29–78)	48.9 (24–73)	0.144
Male gender, *n* (%)	8 (53.3)	10 (66.7)	0.71
Race			
Caucasian, *n* (%)	15 (100)	13 (86.7)	0.483
African, *n* (%)	0 (0)	1 (6.7)	1.0
Asian, *n* (%)	0 (0)	1 (6.7)	1.0
Estimated duration of renal findings (months)	25.2 (0.5–228)	2.8 (0–18)	0.188
S-creatinine µmol/L, (≤100 male, ≤90 female) ^1^	183 (49–464)	246 (85–1065)	0.126
eGFR mL/min/1.73 m^2^, (>60)	50.3 (9–117)	28.3 (4–62)	0.067
eGFR < 60 mL/min/1.73 m^2^, *n* (%)	10 (66.7)	14 (93.3)	0.169
Proteinuria detected (any method), *n* (%)	13/14 * (92.9)	14/14 * (100)	1.0
Hematuria detected (any method), *n* (%)	10/14 * (71.4)	9/14 * (64.3)	1.0
Family history of			
HUS or thrombotic microangiopathy, *n* (%)	0 (0)	1 (6.7)	1.0
kidney disease of unknown origin, *n* (%)	1 (6.7)	3 (20)	0.598
Presented with extrarenal manifestation, *n* (%) ^2^	2 (13.3)	5 (33.3)	0.39
Biopsy indication, *n* (%)			
Acute kidney injury and proteinuria, *n* (%)	1 (6.7)	4 (26.7)	0.33
Acute kidney injury, proteinuria and hematuria, *n* (%)	9 (60.0)	9 (60.0)	1.0
Kidney failure and proteinuria, *n* (%)	2 (13.3)	0 (0)	0.483
Hematuria and proteinuria, *n* (%)	2 (13.3)	0 (0)	0.483
Other, *n* (%) ^3^	1 (6.7)	2 (13.3)	1.0
Clinical diagnosis by study investigators			
aHUS, *n* (%)	5 (33.3)	4 (26.7)	1.0
secondary HUS, *n* (%)	10 (66.7)	11 (73.3)	1.0
Clinical diagnosis by a treating physician			
aHUS, *n* (%)	2 (13.3)	4 (26.7)	0.651
secondary HUS, *n* (%)	5 (33.3)	7 (46.7)	0.71
TMA not otherwise specified, *n* (%)	2 (13.3)	1 (6.7)	1.0
Other, *n* (%) ^4^	6 (40.0)	3 (20.0)	0.427

eGFR = estimated glomerular filtration rate, calculated according to the Chronic Kidney Disease Epidemiology Collaboration (CKD-EPI) equation. ^1^ median (range), ^2^ neurologic symptoms, heart failure, skin symptoms, peripheric gangrene, necrotic lesions of the liver and brain infarction, neuroretinitis, hypertensive retinopathy, ^3^ acute renal insufficiency, or proteinuria, or chronic renal insufficiency with hematuria and proteinuria, ^4^ membranoproliferative glomerulonephritis, systemic lupus erythematosus, mesangial glomerulonephritis NAS, hypertension and postinfectious glomerulonephritis. * Analysis available on the indicated number of patients.

**Table 2 jcm-11-03124-t002:** Factors predisposing to atypical hemolytic-uremic syndrome (aHUS) and potentially causative factors for secondary HUS among patients with histological TMA (h-TMA) and clinical TMA (c-TMA). Values are expressed as numbers of patients (percentages in parenthesis). *p*-values were not significant.

Variable	h-TMA *n* = 15	c-TMA *n* = 15
Predisposing factors identified in aHUS		
History of recent surgery, *n* (%)	0 (0)	1 (6.7)
Histology of recent infection, *n* (%)	0 (0)	2 (13.3)
Multiple factors, *n* (%) ^1^	0 (0)	1 (6.7)
Secondary HUS caused by		
hypertension, *n* (%)	2 (13.3)	3 (20)
History of autoimmune disease, *n* (%)	1 (6.7)	0 (0)
Other, *n* (%) ^2^	1 (6.7)	4 (26.7)
Multiple conditions present, *n* (%) ^3^	6 (40)	4 (26.7)

^1^ Medication, rheumatic disease, stem cell transplantation ^2^ infection, malignant disease/chemotherapy, medication, ^3^ infection, autoimmune disease, malignant disease/chemotherapy, cobalamin C deficiency, medication, hypertension, organ transplantation, essential thrombocytosis/monoclonal gammopathy of unknown significance. h-TMA patients having aHUS had no identifiable predisposing factors.

**Table 3 jcm-11-03124-t003:** Hemolytic variables at the time of diagnosis. Reference values and units are shown in parenthesis. Values are expressed as means (±SD), or medians (range). *p*-value shows the comparisons between histological TMA (h-TMA) and clinical TMA (c-TMA) patients.

Hemolytic Laboratory Variables at Onset (Reference)	h-TMA *n* = 15	c-TMA *n* = 15	*p*-Value
Hemoglobin (134–167 M, 117–155 F, g/L)	119.5 (±14)	103.5 (±17.6)	0.01
Platelets (150–360 E9/L) ^1^	171 (111–503)	187 (72–353)	0.852
LDH (115–235 U/L)	206.9 (±40.4) (*n* = 7 ^2^)	348.4 (±156.2)	0.004
Haptoglobin (0.29–2 g/L) ^1^	1.4 (0.68–1.91) (*n* = 8 ^2^)	0.6 (0–3.42)	0.129
E-Schistocytes (%) (<1)	0.1 (±0.2) (*n* = 6 ^2^)	2 (±1.5) (*n* = 11 ^2^)	0.002

M = male, F = female, LDH = lactate dehydrogenase. ^1^ median (range), ^2^ Analysis available on the indicated number of patients.

**Table 4 jcm-11-03124-t004:** Complement and paraprotein evaluation at the time of diagnosis. Reference values and units are shown in parenthesis. Values are expressed as means (±SD), or number of patients (percentages). *p*-values were not significant.

Baseline Variable (Reference)	h-TMA *n* = 15	c-TMA *n* = 15
Complement proteins		
S-C3 (0.5–1.5 g/L)	1.0 (±0.3) (*n* = 11 *)	1.0 (±0.3) (*n* = 14 *)
Below normal, *n* (%)	0/11 * (0)	1/14 * (7.1)
S-C4 (0.12–0.42 g/L)	0.2 (±0.1) (*n* = 11 *)	0.2 (±0.1) (*n* = 14 *)
Below normal, *n* (%)	1/11 * (9.1)	2/14 * (14.3)
Functional complement analyses		
S-CH100Al (>39%)	97.7 (±48.1) (*n* = 3 *)	101 (±18.5) (*n* = 9 *)
Below normal, *n* (%)	0/3 * (0)	0/9 * (0)
S-CH100C1 (>74%)	104.3 (±29.9) (*n* = 3 *)	96 (±22.5) (*n* = 9 *)
Below normal, *n* (%)	1/3 * (33.3)	2/9 * (22.2)
S-CH100L (>10%)	109.7 (±94.9) (*n* = 3 *)	69 (±53.7) (*n* = 9 *)
Below, *n* (%)	1/3 * (33.3)	2/9 * (22.2)
Complement autoantibodies and mutations		
C3nef positivity, *n* (%)	0/3 * (0)	3/6 * (50)
Factor H antibody positivity, *n* (%)	1/3 * (33.3)	0/5 * (0)
No mutation found, *n* (%)	2/3 * (66.7)	2/5 * (40)
Disease causing mutation found, *n* (%)	0 (0)	1/5 (20)
Only risk haplotype found, *n* (%)	1/3 * (33.3)	0 (0)
Mutation + only risk haplotype found, *n* (%)	0 (0)	1/5 * (20)
Mutation + variant of unknown significance, *n* (%)	0 (0)	1/5 * (20)
Paraprotein finding		
Only in blood, *n* (%)	1/4 * (25)	2/3 * (66.7)
Only in urine, *n* (%)	0/4 * (0)	0/3 * (0)
Both in blood and urine, *n* (%)	2/4 * (50)	0/3 * (0)
Type of paraprotein in serum		
IgG kappa, *n* (%)	1/3 * (33.3)	1/2 * (50)
IgG lambda, *n* (%)	1/3 * (33.3)	1/2 * (50)
Multiple types, *n* (%)	1/3 * (33.3)	0/2 * (0)
Hematologic disease when paraprotein observed		
MGRS or MGUS, *n* (%)	2/3 * (66.7)	2/2 * (100)
Overt malignancy, *n* (%)	1/3 * (33.3)	0/2 * (0)

C3 = complement 3, C4 = complement 4, CH100Al = activity of the alternative pathway of complement, CH100Cl = activity of the classical pathway of complement, CH100L = activity of the lectin pathway of complement, C3nef = complement 3 nephritic factor, MGRS = monoclonal gammopathy of renal significance, MGUS = monoclonal gammopathy of unknown significance. * Analysis available on the indicated number of patients.

**Table 5 jcm-11-03124-t005:** Histological variables of TMA patients. Values are expressed as means (±SD), or number of patients (percentages). *p*-value shows the comparison between histological TMA (h-TMA) and clinical TMA (c-TMA) patients.

Variable	h-TMA *n* = 15	c-TMA *n* = 15	*p*-Value
Glomerular changes			
Capillary wrinkling, *n* (%)	8/14 ^1^ (57.1)	11 (73.3)	0.450
Mesangiolysis, *n* (%)	4 (26.7)	2 (13.3)	0.651
Microaneurysm, *n* (%)	1 (6.7)	1 (6.7)	1.000
Thrombi, *n* (%)	4 (26.7)	4 (26.7)	0.651
Biopsies with crescents, *n* (%)	3 (20)	1 (6.7)	0.598
Sclerotic glomeruli, *n* (%) (SD)	19.5 (±21.6)	13.6 (±15.8)	0.408
Mesangial matrix expansion			
None, *n* (%)	5 (33.3)	9 (60)	0.272
Mild, *n* (%)	6 (40)	3 (20)	0.427
Moderate, *n* (%)	1 (6.7)	0 (0)	1.000
Severe, *n* (%)	3 (20)	3 (20)	1.000
Double basement membrane			
None, *n* (%)	3 (20)	5 (33.3)	0.682
Mild, *n* (%)	5 (33.3)	1 (6.7)	0.169
Moderate, *n* (%)	2 (13.3)	4 (26.7)	0.651
Severe, *n* (%)	5 (33.3)	5 (33.3)	1.000
Lobulated glomeruli, *n* (%)	6 (40)	2 (13.3)	0.215
Tubulointerstitial changes			
Interstitial fibrosis			
None, *n* (%)	6 (40)	11 (73.3)	0.139
Mild, *n* (%)	6 (40)	3 (20)	0.427
Moderate, *n* (%)	2 (13.3)	1 (6.7)	1.000
Severe, *n* (%)	1 (6.7)	0 (0)	1.000
Tubular atrophy			
None, *n* (%)	6 (40)	4 (26.7)	0.700
Mild, *n* (%)	7 (46.7)	9 (60)	0.715
Moderate, *n* (%)	0 (0)	2 (13.3)	0.483
Severe, *n* (%)	2 (13.3)	0 (0)	0.483
Total interstitial inflammation			
None, *n* (%)	7 (46.7)	7 (46.7)	1.000
Mild, *n* (%)	5 (33.3)	7 (46.7)	0.710
Moderate, *n* (%)	3 (20)	1 (6.7)	0.598
Severe, *n* (%)	0 (0)	0 (0)	1.000
Vascular changes			
Myxoid swelling, *n* (%)	1 (6.7)	9 (60)	0.005
Intramural fibrin, *n* (%)	0 (0)	3 (20)	0.224
Thrombi, *n* (%)	0 (0)	3 (20)	0.224
Onion-skinning, *n* (%)	0 (0)	6 (40)	0.017
Arterial sclerosis			
None, *n* (%)	9 (60)	3/14 ^1^ (21.4)	0.060
Mild, *n* (%)	3 (20)	7/14 ^1^ (50)	0.128
Moderate, *n* (%)	3 (20)	2/14 ^1^ (14.3)	1.000
Severe, *n* (%)	0 (0)	2/14 ^1^ (14.3)	0.224
Electron microscopy changes			
Glomerular subendothelial flocculent material, *n* (%)	9/11 ^1^ (81.8)	12/13 ^1^ (92.3)	0.576
Wrinkled glomerular basement membrane, *n* (%)	5/12 ^1^ (41.7)	7/13 ^1^ (53.8)	0.695
Loss of endothelial fenestration, *n* (%)	8/11 ^1^ (72.7)	13/13 ^1^ (100)	0.082
Degree of podocyte effacement,% (SD)	21.4 (±17.0) (*n* = 11 ^1^)	32.7 (±33.1), (*n* = 13 ^1^)	0.295
Doubled glomerular basement membrane, *n* (%)	7/11 ^1^ (63.6)	9/13 ^1^ (69.2)	1.000
Thin basement membrane (<264 nm), *n* (%)	2/10 ^1^ (20.0)	2/13 ^1^ (15.4)	1.000

^1^ Analysis available on the indicated number of patients.

**Table 6 jcm-11-03124-t006:** Longitudinal changes from last follow-up. Reference values and units are given where appropriate. *p*-values were not significant.

Variable	h-TMA *n* = 15	c-TMA *n* = 15
Mean follow-up time since biopsy (years) (range)	6.9 (2.8–12)	5.6 (1.5–13)
On chronic dialysis, *n* (%)	4 (26.7)	5 (33.3)
Kidney transplantation, *n* (%)	2 (13.3)	1 (6.7)
Kidney failure * not necessitating chronic dialysis or transplant, *n* (%)	6 (40.0)	6 (40.0)
Death during follow-up, *n* (%)	4 (26.7)	1 (6.7)
Multiple outbreaks during follow-up, *n* (%)	1 (6.7)	2 (13.3)

* eGFR < 60 mL/min/1.73 m^2^ at last follow-up.

## Data Availability

The data that support the findings of this study are available from the corresponding author, M.K., upon reasonable request. The data that support the findings of this study are not publicly available due to ethical reasons and due to privacy reasons of research participants. Data availability can be requested from the corresponding author, M.K., upon reasonable request.

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
