# Peer review of "Vascular Occlusion in Kidney Biopsy Is Characteristic of Clinically Manifesting Thrombotic Microangiopathy"

_jcm, 2022, doi:10.3390/jcm11113124_

Round 1

Reviewer 1 Report

Dear editors,

The study titled “Vascular occlusion in kidney biopsy is characteristic of clinically manifesting thrombotic microangiopathy” offers information in regard to patients presenting with TMA, highlighting a very important group, which are the clinically inapparent patients. The authors analyzed histopathological features of clinically vs only histologically proven TMA. This is a nice study; however, some improvements must be done. Here, some of my observations:  

Introduction:

  1. In the fourth paragraph (lines 66 – 78) of the introduction, authors refer to secondary TMA. I would be more specific and call it secondary aHUS, as secondary TMA can be any entity (including STEC-HUS or TTP), which are not usually triggered by the numbered factors. For further details use your own reference n°2.
  2. Overall, the fourth paragraph (lines 66 – 78) is confusing, especially in combination with the classification of TMA described in the following paragraph (lines 79 – 86). It could be deleted. In addition, there are some clinical relevant differences between TTP and aHUS which could be specified.  

Methods:

  1. The writing is a bit confusing, dividing the methods into sections would be more helpful (e.g. study population, laboratory analysis, pathological analysis, statistical analysis), especially as this part is very extensive.
  2. Study population: Were pediatric patients also included? Age is not described in the inclusion/exclusion criteria.
  3. There is no description of primary outcome, secondary outcomes in the methods part. This could be specified and give more clarity to what was actually looked for.
  4. Follow up time was not defined. When did it start/end? Does it comprise the time between kidney biopsy until this analysis? This must be clarified.
  5. The significance level of the statistical test was not defined.

Results:

  1. Figure 1 is too small and impossible to evaluate.
  2. In the text it should be specified when the overall group is meant and not each group.
  3. Table 1: as far as I understood the presence of “the classic triad” was inherent to the group classification (clinical vs histological TMA). It makes no sense to compare this characteristic between the groups, even less to make a statistical test.
  4. Table 1: if groups were compared to evaluate the baseline characteristics, it should be included in the table (i. e. p-values).
  5. Table 1: what does “estimated duration of renal findings“ actually mean? In terms of reduced kidney function exactly? Need for renal replacement therapy? This was quite different between groups and should be clarified.
  6. Table 2: it is interesting that so-called predisposing factors were only found in the c-TMA group and should be mentioned.
  7. Table 2 must be reformatted (secondary HUS caused by…)
  8. It is doubtful that the results of the multivariate logistic regression will lead to valid inferences as the sample size is very small. 

Discussion:

  1. It would be interesting to highlight and discuss the fact that triggering factors were only observed in c-TMA vs h-TMA.
  2. In the discussion authors report that TMA patients were biopsied after 12 weeks from onset of symptoms: is this the mean of all 30 patients? Is there any difference in timing of the biopsy between c-TMA and h-TMA groups? And can this explain also the differences observed in the histological patterns (e.g. vascular occlusive changes). This might be relevant information that should be added to the results and further discussed.
  3. If the authors refer to the vascular occlusive changes in the histopathological results as the leading cause for acute kidney impairment, then why do patients with h-TMA also have relevant kidney function impairment later on? Is there any histological correlation to this finding?

Overall the authors provide novel important information of a rare disease. The single-center, retrospective nature of the study however, does limit the generalizability of the results and should be mentioned in the discussion. 

Reviewer 2 Report

Dear Authors  

 Attention to figures 1 and 2  that are disfeatured.  

Your paper on  " Vascular occlusion in kidney biopsy is characteristic of clinically manifesting thrombotic microangiopathy  " is very agreeable to read and propose an interesting and yet unresolved question on the value of tissue kidney markers and clinically manifesting thrombotic microangiopathy  ( TMA ).
This is clinically relevant moreover as we know that some TMA tissue markers are present in patients without clinical manifestations of TMA .
Your paper is well designed and the results are in accordance with TMA physiopathology.
It is important to mention that TMA manifestations are very important nowadays in clinical syndromes of COVID 19 infections besides oncologic diseases and as adverse effects of some drugs.

Reviewer 3 Report

Although a rare entity, TMA is an important nephrological entity due to its significant impact upon patinets' overall and kidney prognosis. Due to its rarity, there are scarce data regarding the differences between hTMA and cTMA. For this reason, the research is of high importance for nephrology medical team. I congratulate the authors for the research and for the quality of the paper. I suggest an English spell check and grammar check, since there are few mistakes in the paper. Also, all the iamges and graphs need to be improved since they are very difficult to be read (and the first one even it is impossibel to be read). I also suggest to remove the "laboratory indicators of hemolysis" from line 22, since their presence is one that makes the diffecrence between the hTMA and cTMA.
